# Identification of a Panel of miRNAs Associated with Resistance to Palbociclib and Endocrine Therapy

**DOI:** 10.3390/ijms25031498

**Published:** 2024-01-25

**Authors:** Rosalba Torrisi, Valentina Vaira, Laura Giordano, Bethania Fernandes, Giuseppe Saltalamacchia, Raffaella Palumbo, Carlo Carnaghi, Vera Basilico, Francesco Gentile, Giovanna Masci, Rita De Sanctis, Armando Santoro

**Affiliations:** 1Medical Oncology and Hematology Unit, Cancer Center, IRCCS Humanitas Research Hospital, Rozzano, 20089 Milano, Italy; giuseppe.saltalamacchia@humanitas.it (G.S.); giovanna.masci@humanitas.it (G.M.); rita.de_sanctis@hunimed.eu (R.D.S.); armando.santoro@cancercenter.humanitas.it (A.S.); 2Division of Pathology, Fondazione IRCCS Ca’ Granda-Ospedale Maggiore Policlinico, 20122 Milano, Italy; valentina.vaira@unimi.it (V.V.); francesco.gentile0992@gmail.com (F.G.); 3Department of Pathophysiology and Transplantation, University of Milan, 20122 Milano, Italy; 4Biostatistic Unit, IRCCS Humanitas Research Hospital, Rozzano, 20089 Milano, Italy; laura.giordano@humanitas.it; 5Pathology Department, IRCCS Humanitas Research Hospital, Rozzano, 20089 Milano, Italy; bethania.fernandes@humanitas.it; 6Oncologia Medica IRCCS ICS Maugeri, 27100 Pavia, Italy; raffaella.palumbo@icsmaugeri.it; 7Clinical Trials Unit, Istituto Clinico Humanitas, Centro Catanese di Oncologia, 20072 Catania, Italy; carlo.carnaghi@humanitascatania.it; 8Medical Oncology Unit, Istituto Clinico Mater Domini Humanitas, Castellanza, 21100 Varese, Italy; vera.basilico@materdomini.it; 9Department of Biomedical Sciences, Humanitas University, 20072 Pieve Emanuele, Italy

**Keywords:** miRNAs, hormone receptor positive/HER2 negative, metastatic breast cancer, palbociclib, sTILs

## Abstract

We investigated whether we could identify a panel of miRNAs associated with response to treatment in tumor tissues of patients with Hormone Receptor-positive/HER2-negative metastatic breast cancer treated with endocrine therapy (ET) and the CDK4/6 inhibitor (CDK4/6i)i palbociclib. In total, 52 patients were evaluated, with 41 receiving treatment as the first line. The overall median PFS was 20.8 months (range 2.5–66.6). In total, 23% of patients experienced early progression (<6 months). Seven miRNAs (*miR-378e*, *miR-1233*, *miR-99b-5p*, *miR-1260b*, *miR-448*, *-miR-1252-5p*, *miR-324-3p*, *miR-1233-3p*) showed a statistically significant negative association with PFS. When we considered PFS < 6 months, *miR-378e*, *miR-99b-5p*, *miR-877-5p*, *miR-1297*, *miR-455-5p*, and *miR-4536-5p* were statistically associated with a poor outcome. In the multivariate analysis, the first three miRNAs confirmed a significant and independent impact on PFS. The literature data and bioinformatic tools provide an underlying molecular rationale for most of these miRNAs, mainly involving the PI3K/AKT/mTOR pathway and cell-cycle machinery as cyclin D1, CDKN1B, and protein p27^Kip1^ and autophagy. Our findings propose a novel panel of miRNAs associated with a higher likelihood of early progression in patients treated with ET and Palbociclib and may contribute to shed some light on the mechanisms of de novo resistance to CDK4/6i, but this should be considered exploratory and evaluated in larger cohorts.

## 1. Introduction

About 70% of breast cancers are hormone receptor-positive HER2-negative (HR+/HER2−) [1]. The combination of endocrine agents and CDK4/6 inhibitors (CDK 4/6i) has been established as standard treatment for HR+/HER2− metastatic breast cancer (MBC), having consistently doubled progression free survival (PFS) and prolonged overall survival (OS) in some studies [2,3,4,5,6,7,8]. However, about 20% of patients do not derive any benefit from treatment [2,3,4,5,6,7].

Multiple mechanisms of resistance to CDK4/6i have been proposed involving either cell-cycle specific or non-specific resistance [9]. Despite this increased knowledge, no definite clinical, pathological, or molecular predictive factor of sensitivity or resistance has been identified yet.

MicroRNAs (miRNAs) are endogenous, non-coding 19–22-nucleotide-long RNAs that mediate a posttranscriptional negative regulation of gene expression. A single miRNA molecule can target multiple mRNAs and, conversely, one mRNA can be the target of multiple miRNAs [10]. Consequently, miRNAs regulate multiple cellular processes, including cell proliferation, migration, and apoptosis [10]. Deregulated expression of miRNAs is frequently linked to tumor progression [10]. Moreover, miRNAs are emerging as novel potential predictive/prognostic biomarkers of disease and response to therapies [11].

A recent systematic review of miRNAs associated with response to CDK4/6i in solid tumors and hematological malignancies identified six miRNAs (*miR-126*, *miR-326*, *miR-3613-3p*, *miR-29b-3p*, *miR-497*, and *miR-17-92*) associated with response and six miRNAs (*miR-193b*, *miR-432-5p*, *miR-200a*, *miR-223*, *Let-7a*, and *miR-21*) which conferred resistance to treatment [12]. In addition, other miRNAs (*miR-124a*, *miR-9*, *miR-200b*, and *miR-106b*) were shown to mediate cellular response to CDK4/6i without affecting sensitivity to treatment. However, only *miR-432*, *miR-223*, *miR-3613-3p*, and *miR-29b-3p* were investigated in breast cancer tumor tissues [12]. In particular, *miR-432-5p* was associated with resistance to Palbociclib either in breast cancer cell lines (BCCL) and breast tumor biopsies [13].

Stromal tumor-infiltrating lymphocytes (sTILs), defined as the percentage of lymphocytes in the tumor stromal area, are the most reproducible immune parameter scored by pathologists [14,15]. No consistent data on the potential clinical relevance of sTILs or other immune biomarkers are available in HR+/HER2− MBC [16]. Higher basal levels of sTILs have been associated with better response to neoadjuvant chemotherapy also in HR+HER2− tumors, but no predictive biomarker is available for target therapies in this tumor subset either in the early or advanced setting [17].

In the present study, we aimed to identify a panel of miRNAs assessed in tumor tissues which were associated with response to the combination of endocrine therapy (ET) and CDK4/6i palbociclib in patients with HR+/HER2−MBC. The presence of sTILs and their association with outcome was also investigated to show whether the immune microenvironment might be associated with the activity of the CDK4/6i.

## 2. Results

We retrieved from patient files of the four participating institutions patients who fulfilled eligibility criteria for enrollment in the study.

Overall, we obtained tumor tissues from 67 patients fulfilling the eligibility criteria. Twelve cases were discharged because of low RNA quantity. In addition, a further three samples were discharged due to miRNA counts (parameter: probes above the threshold) less than 50%. On the other hand, two samples were not available for sTILs assessment.

Finally, miRNAs and sTILs were determined in 52 and 50 samples, respectively. Tissue samples were obtained from the breast (11), visceral sites (21, of which 13 were from liver), and from soft tissues (20, of which 13 were from lymph nodes). In total, 51 samples were obtained from women and 1 sample was obtained from a male.

Since sTILs were detected only in 15 samples (30%), at a lower than expected rate for HR+/HER2– tumors, we did not perform a subpopulation analysis of immune components but just investigated the association of sTILs’ presence in tumor tissue and patient outcome and with clinical and other pathological features.

The patient and tumor characteristics are summarized in Table 1.

Most patients were treated as first line (41 out of 52), of whom 27 received letrozole and 14 Fulvestrant; only 1 patient received letrozole in second and third line, while 6 and 3 patients received Fulvestrant in second and later lines, respectively.

The overall median PFS (mPFS) was 20.8 months (range 2.5–66.6); the 18-month progression-free proportion was 51.9% (25 patients), and 24-month progression-free proportion was 40.4% (21 patients). The PFS of relevant subgroups is reported in Table 2. A statistically significant longer PFS was observed in patients treated with letrozole, patients with Luminal A tumors, those treated in first line, and those who were endocrine sensitive and endocrine naïve as compared to those endocrine resistant (Table 2).

Patients with ECOG PS = 0 had a longer PFS (25.4 months) than patients with a PS ≥ 1 (12.6 months), with a *p*-value close to statistical significance, *p* = 0.056, but no association between reduced dose and outcome was observed.

No other clinical and pathological feature was significantly associated with PFS. Patients with visceral metastases had a non-statistically significantly shorter PFS as compared to those without visceral metastases, while no difference was observed in patients without and with liver metastases: mPFS = 20.8 and 19.4, respectively.

Twelve out of fifty-two patients (23%) experienced early progression (<6 months), among this subgroup, eight patients were treated with Fulvestrant and four with letrozole; seven patients were treated as first line and the remaining were treated as later lines.

Patients with sTILs containing tumors had a nearly double PFS (36 vs. 19 months) as compared with that of patients with sTILs negative tumors, although the difference was not statistically significant, possibly due to the small sample size. No clinical or pathological feature was associated with the presence of sTILs. Moreover, sTILs-positive tumors had a high, albeit non-statistically significant, positive association with 24-month PFS (OR = 3.27, CI 95%: 0.93; 11.49, *p*-value: 0.064).

### miRNA Analyses

Table 3 summarizes miRNAs which were associated with PFS in univariate analysis considering a *p*-value < 0.1.

Seven miRNAs, namely *miR-378e*, *miR-1233-3p*, *miR-99b-5p*, *miR-1260b*, *miR-448*, *miR-1252-5p*, and *miR-324-3p*, showed a statistically significant negative association with PFS. *MiR-378e*, *miR.99b-5p*, *miR-1260b*, and *miR-216a-5p* showed also a statistically significant negative association with the likelihood of being progression-free at 18 months (Table 3).

On the other hand, only *miR-196a-5p* and *miR-342-5p* were positively associated with the likelihood of being progression-free at 18 months (Table 3), while only first line treatment and *miR-376a-3p* were significantly and positively associated with the likelihood of being progression free at 24 months (OR: 0.105, 95%CI: 0.1;0.9, *p*-value = 0.039 and OR: 1.007, 95%CI: 1.00; 1.01, *p* = 0.037, respectively).

We then performed a multivariate analysis including the miRNAs and clinical variables that were statistically significantly associated with PFS in univariate analysis (endocrine agent, tumor subtype, and ECOG PS) (Table 4).

Each of the miRNAs included maintained a significant independent association with the outcome adjusted for clinical variables, but due to the high correlation between miRNAs, they were not included at the same time (Table 4). The estimates reported in the table for miRNAs are those obtained when each mRNA was individually included in the model with clinical variables.

When we considered PFS shorter than 6 months as an outcome, the line of therapy (first vs. later), ECOG PS (0 vs. ≥1), and previous ET were statistically and positively associated with the likelihood of progressing within 6 months, as were *miR-378e*, *miR-99b-5p*, *miR-877-5p*, *miR-1297*, *miR-455-5p*, and *miR-4536-5p* (Table 3). In the multivariate analysis, only *miR-378e* ((OR: 1.05, 95%CI: 1.01; 1.09, *p* value 0.002 unadjusted), *miR-99b-5p* (OR: 1.01, 95%CI: 1.00; 1.02, *p* value 0.022 adjusted for ECOG PS), and *miR-877-5p* (OR: 1.07, 95%CI: 1.01; 1.15, *p* value 0.032 adjusted for line of therapy) confirmed a significant and independent impact on PFS.

To obtain preliminary insights of the potentially involved pathways, we searched for predicted targets of *miR-99b-5p*, *miR-378e*, and *miR-877-5p*, the three miRNAs associated with early progression (Appendix A). A functional annotation of the predicted targets using STRING database and the Gene Ontology, KEGG, and Reactome functions (Figure 1A,B and Appendix A) showed that the predicted genes are involved in cell catabolic processes (translation regulation, autophagy, and mitophagy; Figure 1A,B), the cell cycle, and endocrine resistance (Figure 1A,B).

To corroborate these data, we performed an enrichment analysis of predicted targets using the overrepresentation tool (ORA; Appendix A). The results from ORA confirmed that cell metabolism (mTOR/Akt signaling, Skeletal muscle hypertrophy), senescence, and endocrine resistance are potentially affected by the three miRNAs (Appendix A). Further, ORA suggested that the cell cycle is also potentially regulated by the three miRNAs (highlighted in grey in Appendix A), suggesting that they could be used as markers of CDKi therapy resistance.

## 3. Discussion

Even though the advent of CDK4/6i has dramatically improved the outcomes of patients with HR+/HER2− advanced breast cancer, about 20% of patients do not benefit from this therapy and may require different approaches upfront [2,3,4,5,6,7].

Mechanisms of resistance to CDK4/6i have been extensively studied in in vitro and in vivo models and include cell cycle-specific resistance (as Rb loss, E2F amplification, overexpression of tumor suppressor factors, CDK amplification) and cell cycle-nonspecific resistance (as activation of Fibroblast growth factor receptor -1, PI3K/AKT/mTOR, MAPKinase signaling pathways, loss of ER and PgR, overexpression of AR) [9].

Recently, some miRNAs have also been associated with CDK4/6i [12]. Cornell et al. have studied an mRNA profile (30 miRNAs) of the parental- and palbociclib-resistant T47D cell line and focused on five miRNAs which showed a >100-fold difference, and on *miR-432-5p*, whose expression was associated with increased levels of CDK6, a putative marker of resistance to CDK 4/6i [13]. Parental cells overexpressing *miR-432-5p* behaved much like palbociclib-resistant cells and, furthermore, when resistant T47D and MCF7 cells were transfected with an *miR-432-5p* inhibitor, CDK6 levels decreased and G1 arrest increased [12]. Moreover, in 44 biopsies obtained from patients treated with another CDK4/6i ribociclb, *miR-432-5p* and CDK6 expression were higher in both intrinsically and acquired resistant tumors as compared with biopsies obtained in patients with sensitive disease [13]. Also, miR-223 has been associated with resistance to palbociclib, whereas a positive association with response to the drug was found for *miR-3613-3p* and *miR-29b-3p* in breast cancer tissues [12].

Our results, in a relatively small cohort of patients treated with ET and Palbociclib mostly as first line, showed a panel of miRNAs which were mainly negatively associated with outcome independently of predictable clinical variables such as tumor subtype and endocrine agent. More interestingly, we also identified a panel of miRNAs which were significantly associated with an increased likelihood of early progression (<6 months). In our study, we featured 23% of patients who were early progressing, a figure which is consistent with the proportion of approximately 20% of patients progressing within 6 months in first line pivotal clinical trials of CDK4/6i plus ET [2,3,4,5,6,7].

The identification of biomarkers associated with the likelihood to rapidly progress on first line therapy is of utmost importance to provide timely alternative and more effective treatments. A large amount of data on biomarkers associated with early progression with ET and CDK4/6i have been published, but no conclusive evidence is available [reviewed in [18]].

An analysis of circulating markers of early progression in patients treated with Fulvestrant and palbociclib/placebo in the PALOMA 3 trial showed that circulating tumor fraction, TP53 mutation, and FGFR1 gain were each independently associated with risk of early relapse for both Fulvestrant alone and Fulvestrant plus palbociclib subgroups [19]. The population treated within the PALOMA 3 trial included all pretreated patients, with 33% and 40% receiving treatment after chemotherapy or after two lines of ET, respectively; thus, the mechanism of resistance may be different from those developing in patients currently treated with CDK4/6i mostly as first-or second-line treatment [20].

In support of this speculation, alterations of FGFR1 were associated with early progression also in the MOnaLEEsa2 and in BioItaLEE trials with ribociclib but not in the MonaLEEsa 3 trial with the same drug, and TP53 mutations were associated with ribociclib only in BioItaLEE [20]. On the other hand, some but not all the neoadjuvant and first-line studies with CDK4/6i have suggested a role for the upregulation of CCNE1 as a biomarker of early resistance [18].

In our study, we identified three miRNAs, *miR-378e*, *miR-99b-5p*, and *miR-877-5p*, which, independently of clinical variables such as line of therapy, ECOG PS, and previous ET, predicted a very poor outcome, indicating a de novo resistance to CDK4/6i.

The *miR-99* family which includes *miR-99b-5p* is implicated in virtually all known human cancers, either promoting (oncogenic miRNA, oncomiR) or suppressing (TSmiR) tumor growth or, as occurs in breast cancer, with both functions [21]. The Akt/mTOR pathway, which is one of the most recognized mechanisms of resistance to ET plus CDKi, is among the preferred targets of *miR-99b-5p* [21]. The literature data are inconsistent. An inverse relationship between the expression of *miR-99b-5p* and mTOR has been described either in BCCL and tumor specimens [22]. Oppositely, an analysis of two large datasets of breast cancer patients consisting of 1961 cases downloaded from the METABRIC and TCGA databases showed that high *miR-99b* expression correlated significantly with enriched mTORC1 gene sets and that the mTOR pathway, but no other signaling pathway, was activated in *miR-99b*-high-expressing breast cancer specimens [23]. In the same datasets, *miR-99b*-high breast cancer specimens were significantly enriched in three genes related to cell proliferation: E2F targets, the G2/M checkpoint, and mitotic spindle gene sets. Importantly, E2F is a downstream effector of CDK4 and CDK6 on the cell cycle and its overexpression may hamper the activity of CDK4/6i. Moreover, high *miR-99b* expression was associated with worse patient survival, particularly in HR+/HER2− tumors [23]. Our results seem to support an oncomiR function for this miRNA in patients treated with ET and CDK4/6i.

A not univocal function is presented also by *miR-877-5p* which was suppressed in breast cancer tissues but, on the other hand, promoted bone metastasis in in vitro and in vivo models [24,25]. In our study, an independent association with resistance to ET plus palbociclib was shown. A possible explanation for its oncomiR function in this setting might derive from non-cancer models such as ARDS, where *miR-877-5p* suppressed CDKN1B which encodes p27^Kip1^, whose downregulation is claimed as one of the possible mechanisms of resistance to CDK4/6i [26]. In addition, CDKN1B inhibited the activation of the PI3K/Akt signaling pathway [26].

As of now, to our knowledge, no direct correlation with breast cancer and *miR-378e* has been reported yet. However, the major predicted target of miRNA is kallikrein-related peptidase (KLK) 4, a serine protease which has been associated with several types of cancer [27]. As for breast cancer, increased expression of *mir-378* and of KLK-4 were independently and negatively correlated with prognosis in triple-negative breast cancer [27].

Bioinformatics tools confirmed a predicted involvement of the three above-mentioned miRNAs and AKT/mTOR signaling, cell-cycle regulation, endocrine resistance, and with cell catabolic processes such as autophagy.

Autophagy is a highly conserved homeostatic process whose predominant role in cancer cells is to confer stress tolerance, maintaining tumor cell survival, but, at the same time, it may also play a role in cancer cell death and tumor suppression [28]. Autophagy can be regulated by miRNA via posttranscriptional regulation of autophagy-related protein expression [29]. Palbociclib has been shown to induce autophagy in other cancer models such as hepatocellular carcinoma and gastric cancer [28]. In breast cancer models, cells activate autophagy in response to Palbociclib, and a blockade of autophagy significantly improved the efficacy of CDK4/6 inhibition in in vitro and in vivo breast cancer models with an intact G1/S transition [30]. Transcriptomic profiling results of palbociclib-sensitive and -resistant breast cancer cells revealed that resistant cells present upregulation of many autophagy-related genes [31]. Since autophagy was one of the predicted gene networks targeted by miRNAs associated with resistance to palbociclib, our findings seem to support this preclinical evidence.

The PI3K/AKT/mTOR pathway is the target through which *miR-1297* also promotes breast cancer [32]. This miRNA was significantly upregulated in breast cancer tissues as compared to normal adjacent tissue and in BCCL as compared to normal mammary epithelial cells. In addition, *miR-1297* was an independent factor for predicting both 5-year OS and PFS. At the molecular level, *miR-1297* promoted cell proliferation, cell cycle progression, and inhibited apoptosis of breast cancer cells at least partially by activating PI3K/AKT signaling by targeting PTEN. *MiR-1297* was also shown to increase cyclin D1, another putative mechanism of resistance to CDK4/6i [32]. Similarly, *miR-455-5p* has been shown to be highly expressed in breast cancer tissues and to act as an independent prognostic factor for OS [33].

The clinical results are in line with those expected in a real-world series of patients mostly treated as first-line therapy with the predictable impact of the tumor subtype and the line of treatment on the outcome. ECOG PS too was strongly associated with outcome but this effect was not due likely to palbociclib dose since no association between reduced drug and outcome was found. Notably, no difference was observed among patients with recurrent vs. de novo MBC, differently from what has been reported both in clinical trials and real-world series [34].

An interesting finding was the doubled mPFS and the 3-fold greater OR for the likelihood of not progressing before 24 months observed in patients with sTILs-positive tumors. No clear evidence on the role of the immune microenviroment in affecting the response to CDK4/6i is available; even though our finding appears intriguing, the small number of patients with sTILs-positive tumors in our study means that further evaluation in larger cohorts are necessary before seeking for a biologic explanation.

The number of patients included in the study compared with the elevated number of miRNAs examined represents a major limitation of our study. However, some of the miRNAs proposed consistently maintained an association with outcome and, importantly, the literature data and bioinformatic tools provide an underlying molecular rationale, mainly involving the PI3K/AKT/mTOR pathway, cell-cycle machinery, in particular cyclin D1, CDKN1B, protein p27^Kip1^, and E2F, and catabolic processes as autophagy which support our results. Another strength of our work is that the tumor tissues were obtained immediately before starting palbociclib, ruling out the interference of previous treatments.

A prospective validation of this panel of miRNAs, possibly assessed in liquid biopsies to overcome the need for metastatic tumor tissue, might represent an initial step to corroborate our findings, as well as in vitro studies. When validated, the identification of a panel of miRNAs predicting de novo resistance to treatment would reduce the use of an ineffective treatment for a not-negligible rate of patients (about 20%), who presumably may not derive any benefit from the standard first-line treatment with ET and CDK 4/6i.

In addition, in the era of mRNA-based vaccines as an anti-cancer strategy (which may be fueled by the positive results of the trial KEYNOTE 942, even though this was in a much more antigenic tumor, such as melanoma) [35], the identification of miRNAs involved in the response to treatment could help to develop specific mRNA-directed vaccines to enhance CDK 4/6i activity.

## 4. Materials and Methods

Patients with HR+/HER2− recurrent breast cancer or MBC who were treated with an aromatase inhibitor or Fulvestrant (+GnRH analogue if premenopausal) and palbociclib as a first- or later line of treatment for advanced disease at the 4 participating Institutions (IRCSS Humanitas Research Hospital—Rozzano, Humanitas MaterDomini—Castellanza, Centro Oncologico Catanese—Catania, and IRCSS Fondazione Salvatore Maugeri—Pavia) were eligible for the study. Patients had undergone a biopsy of the metastatic tissue (or from the primary breast tumor if synchronous metastases were in bone or in not accessible sites) for diagnostic purposes immediately before starting treatment with CDK4/6i and had a follow up of at least one year if not progressed before to be included in the study. Patients with metachronous bone disease only were excluded from the study because of technical issues in miRNA detection from decalcified samples.

The primary objective of the study was to identify a panel of miRNAs which might be associated with PFS in patients treated with palbociclib and ET. Secondary objectives were to determine sTILs levels in the same pretreatment biopsies and correlate them with clinical outcome on CDK 4/6i and ET and to correlate clinical and pathological features of the tumor with the expression of molecular markers (miRNAs and sTILs).

Clinical features analyzed were age, menopausal status, de novo or recurrent MBC, number and type of metastatic sites, line of therapy of the CDK4/6i, type of ET associated with Palbociclib, ECOG Performance Status (PS), dose reduction. In addition, pathological features of tumor biopsy such as high (>50%), low (10–49%) or negative (<10%) PgR expression, tumor proliferation assessed by high (>20%) and low (≤20%) Ki67, HER2 expression (negative and low), and tumor subtype (Luminal A and B defined according to ki67 > 20% and/or PgR negative) were assessed and associated with outcome and molecular markers. Patients who had never received ET were considered as endocrine naïve, while endocrine-sensitive and -resistant patients were defined according to the classical definitions.

### 4.1. miRNA Analyses

Total RNA was purified from tumor-enriched sections of 68 samples obtained from 67 patients using the RecoverAll Total Nucleic Acid Isolation Kit for FFPE (Thermofisher Scientific, Rodano, MI, Italy) followed by gDNA digestion and according to manufacturer’s instruction. RNA quality and quantity was measured with the High-Sensitivity RNA ScreenTape system (Agilent Technologies, Cernusco Sul Naviglio (Mi), Italy). Then, 50 ng of total RNA were used for miRNA profiling using the nCounter Human v3 miRNA Panel (NanoString Technologies, Seattle, WA, USA), as previously described [36]. The nCounter Flex instrument was used and all counts were gathered by scanning for 280 fields of view per sample. Raw data were analyzed using the nSolver software version 2.0.134 (NanoString Technologies). Specifically, the background threshold was set at the highest value of negative controls and any value below this threshold was converted to zero. miRNAs whose counts were negative in up to half of the samples were removed (n = 504; 63%). After that, miRNA counts were normalized using the entire array and normalized miRNA values were subjected to statistical analyses.

Bioinformatics prediction of miR-related signaling was performed using miRTargetLink 2.0 web-based tool for identification of validated targets [37]. For signaling pathway prediction, target gene over-representation analysis (ORA) using the GeneTrail 3 function available within miRTargetLink 2.0, or the STRING database, was used, as previously described [36,37,38]. Briefly, for ORA output, we considered Biocarta and KEGG pathways with an FDR-corrected *p* value less or equal 0.01. For STRING analysis, we used as input the predicted targets and, as reference, the human genome.

### 4.2. sTILs Analyses

sTILs were assessed and scored in FFPE tumor biopsies according to the recommendations of the TILs working group [14]. In particular, sTILs from the stromal compartment, within the borders of the invasive tumor, were reported. All mononuclear cells (including lymphocytes and plasma cells) were scored. Polymorphonuclear leukocytes were excluded. One section (2–3 μm, ×200–400 magnification) per patient was sufficient. sTILs were scored as a categorical variable (present/absent).

### 4.3. Statistical Analyses

Clinical and demographic data and miRNAs were described as number and proportion or as median and range. The objective of this study was to explore the correlation between miRNA expression and PFS considered as the time between starting treatment and progression/death or last contact, whichever occurred first. Progression was also considered as binomial outcome at specific time points considering as event the probability of having disease progression or death before 6 months or after 18 and 24 months.

Survival curves for PFS were generated by the Kaplan–Meier method and differences between groups were compared by the log rank test. The Cox proportional hazards regression model was used to calculate the hazard ratios (HRs) and their 95% confidence intervals (Cis) both in univariate and multivariate analyses. To evaluate the probability for the event as binomial outcome, the logistic regression model was used and odds ratios with their 95% confidence intervals were reported.

The final model was built considering all factors statistically significant at level *p* = 0.10 in the univariable setting and which confirmed their effect in the multivariable model at level *p* = 0.050. In the analysis of the 295 miRNAs, the adjusted α level was equal to 0.0002 considering the Bonferroni Correction. All analyses were performed using SAS v. 9.4 (SAS Institute, Cary, NC, USA).

Considering the sample size and the estimated effect sizes, the power of analyses was not controlled. The results should be considered explorative in nature; the multivariable models are descriptive and should be considered as starting points for further analyses.

## 5. Conclusions

In conclusion, in the present study, we propose a novel panel of miRNAs which predicted a higher likelihood of an early progression in patients treated with endocrine agents and the CDK 4/6i palbociclib. Due to the relatively small number of patients, our findings should be considered as exploratory and hypothesis generating and deserve to be validated in larger cohorts and in in vitro, but they may contribute to shedding some light on the mechanism of de novo resistance to CDK4/6i.

## Figures and Tables

**Figure 1 ijms-25-01498-f001:**
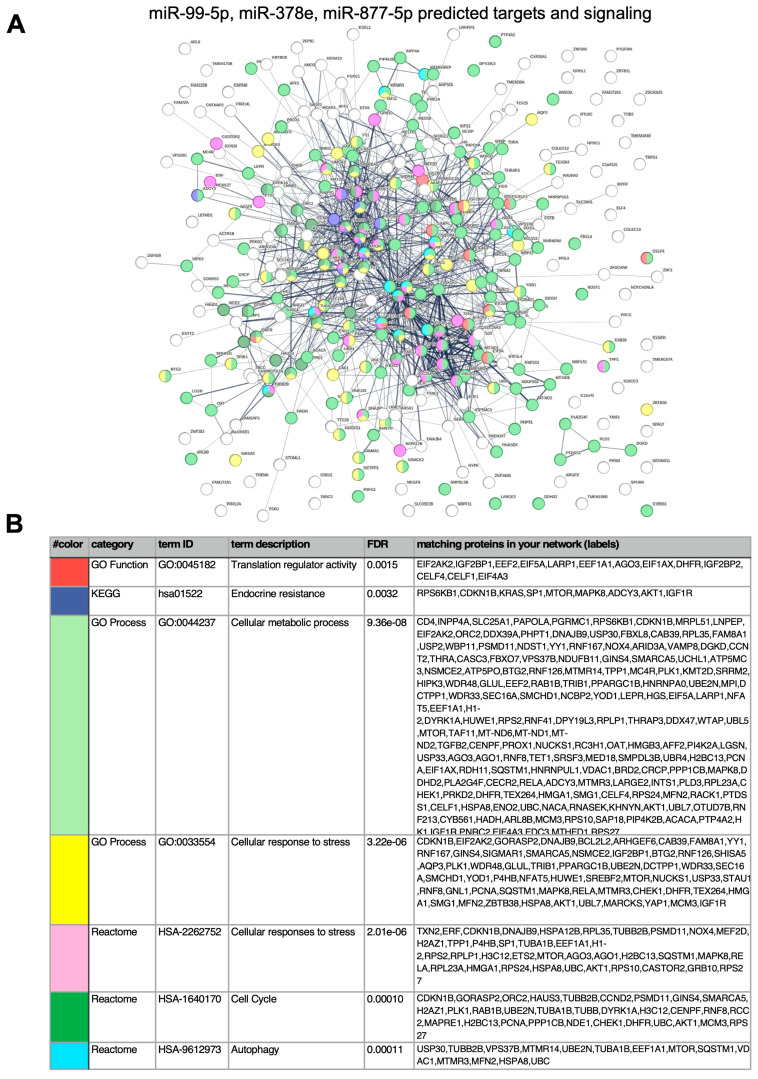
(**A**) STRING analysis with potential targets of the miRNAs miR-99-5p, −378e, and miR-877-5p (listed in Appendix A) was performed and gene networks were searched using Gene Ontology (GO), KEGG, or Reactome tools implemented within STRING. The human genome was used as a reference. (**B**) The full list of identified enriched terms is shown in Appendix A. FDR, false discovery rate *q* value.

**Table 1 ijms-25-01498-t001:** Patient and tumor characteristics.

	N	%
Age in years, median (range)	64.5	(33–84)
Menopausal status		
Pre/peri-menopausal	7	13.5
Post-menopausal	44	84.6
Not applicable	1	1.9
Occurrence of MBC		
De novo	13	25
rMBC	39	75
PgR% (biopsy)		
High (≥50%)	18	34.6
Low (10–49%)	13	25
Negative (<10%)	21	40.4
Ki67% (biopsy)		
≤20	35	67.3
>20	16	30.8
Unknown	1	1.9
Tumor subtype		
Luminal A	27	51.9
Luminal B	25	48.1
Metastatic sites (number)		
≤2	28	53.9
>2	24	46.1
Metastatic sites		
Bone	26	50
Visceral	33	63.5
Other	38	73.1
Endocrine resistance		
Endocrine naïve *	11	21.2
Endocrine sensitive **	23	44.2
Endocrine resistance primary	2	3.8
Endocrine resistance secondary	16	30.8
Prior lines of therapy for metastatic disease		
0	41	78.8
1	6	11.5
≥2	5	9.6
Endocrine therapy		
Letrozole	29	55.8
Fulvestrant	23	44.2
ECOG Performance status		
0	35	67.3
≥1	17	32.7
Dose reduction		
No	27	51.9
yes	24	46.2
unknown	1	1.9

MBC metastatic breast cancer; * endocrine naïve patients who had never received endocrine therapy; ** endocrine sensitive patients who had recurred at least 12 months after completing adjuvant endocrine therapy or had progressed at least after 6 months of previous endocrine therapy for advanced disease.

**Table 2 ijms-25-01498-t002:** Median PFS (mPFS) in relevant subgroups.

	N	mPFS Months (95% CI)	*p*-Value
Overall.	52	20.8 (12.6–21.6)	
Menopausal status, n			
Pre/peri-menopausal	7	25.4 (3.4–NR)	
Post-menopausal	44	20.9 (12.4–36)	NS
Not applicable	1	-	
Occurence of MBC			
De novo	13	23.8 (15.8-NR)	
Recurrent	39	18 (8.7–36.2)	NS

Tumor subtype (biopsy), n			
Luminal A	27	31.6 (15.8–NR)	
Luminal B	25	16.6 (5.1–23.8)	<0.05
Ki67% (biopsy)			
≤20	35	25.4 (12.4–36.2)	
>20	16	18.4 (4.2–28.3)	NS
Unknown	1	-	
Metastatic sites			
Visceral	33	19.4 (8.5–31.6)	
Non visceral	19	36.2 (12.4–NR)	NS
Metastatic sites (number)			
≤2	28	23.8 (15.8–36)	
>2	24	16.8 (5.8–NR)	NS
Endocrine therapy			
Letrozole	29	31.6 (18–NR)	
Fulvestrant	23	8.7 (5.1–19)	<0.005
Prior lines of therapy for metastatic disease			
0	41	25.4 (16.6–36.4)	
≥1	11	8.5 (3.1–17.6)	<0.005
Endocrine resistance			
Endocrine naive *	11	28.3 (15.8–NR)	
Endocrine sensitive **	23	36.2 (12.4–NR)	
Endocrine resistant	16	8.5 (3.5–19)	<0.05
sTILs			
Negative	35	19 (8.7–25.4)	
Positive	15	36 (3.5–NR)	NS
Unknown	2	-	
ECOG Performance Status			
0	35	25.4 (16–36.4)	
≥1	17	12.6 (4.2–20.9)	NS

mPFS: median progression survival; 95% CI: 95% confidence intervals; MBC: metastatic breast cancer; sTILs stromal tumor infiltrating lymphocytes. * endocrine-naïve patients who had never received endocrine therapy; ** endocrine sensitive patients who had recurred at least 12 months after completing adjuvant endocrine therapy or had progressed at least after 6 months of previous endocrine therapy for advanced disease.

**Table 3 ijms-25-01498-t003:** miRNAs associated with outcome with a *p*-value < 0.1.

		PFS			≥18 Months PFS			<6 Months PFS	
**mmirRNA**	**HR**	**95% CI**	***p* Value**	**OR**	**95% CI**	***p* Value**	**OR**	**95% CI**	***p* Value**
miR-216a-5p	1.025	1.000–1.051	0.053	**0.951**	**0.910–0.994**	**0.026**			
**miR-378e**	**1.017**	**1.001–1.032**	**0.034**	**0.961**	**0.929–0.994**	**0.022**	**1.060**	**1.019–1.102**	**0.004**
**miR-1233–3p**	**1.011**	**1.003–1.019**	**0.009**	0.989	0.975–1.002	0.097			
**miR-99b-5p**	**1.004**	**1.001–1.008**	**0.013**	**0.993**	**0.986–1.000**	**0.046**	**1.009**	**1.002–1.017**	**0.014**
**miR-1260b**	**1.002**	**1.001–1.004**	**0.015**	**0.994**	**0.989–1.000**	**0.034**			

**miR-448**	**1.038**	**1.003–1.075**	**0.032**						
**miR-1252–5p**	**1.029**	**1.001–1.058**	**0.041**						
**miR-324–3p**	**1.027**	**1.001–1.053**	**0.04**						
miR-132–3p	1.004	1.000–1.008	0.050						
miR-19a-3p	0.991	0.981–1.001	0.067						

**miR-196a-5p**				**1.001**	**1.000–1.001**	**0.023**			
**miR-342–5p**				**1.057**	**1.001–1.116**	**0.046**			
miR-410–3p				0.902	0.808–1.006	0.064			

miR-3161				1.067	0.944–1.144	0.071			
miR-424–5p				1.002	1.000–1.003	0.070			

**miR-1297**							**1.013**	**1.001–1.026**	**0.041**
**miR-877–5p**							**1.065**	**1.005–1.129**	**0.032**
**miR-4536-5p**							**1.050**	**1.007–1.096**	**0.023**

PFS: progression-free survival; HR: hazard ratio; 95%CI: 95% confidence interval; OR: odd ratio. In bold, miRNAs which were statistically significantly associated with outcome with (*p* < 0.05).

**Table 4 ijms-25-01498-t004:** Multivariable miRNA estimates for PFS adjusted for statistically significant clinical factors (*p* < 0.05).

Variable	Hazard Ratio	95% CI	*p*-Value
ECOG PS (≥1 vs. 0)	3.90	1.40–10.93	0.010
Endocrine agent (Fulvestrant vs. letrozole)	2.94	1.25–6.94	0.014
Tumor subtype (Luminal A vs. B)	0.25	0.10–0.66	0.005
miR-410-3p	1.05	1.00–1.10	0.041
miR-448	1.06	1.00–1.13	0.035
miR-1252-5p	1.03	1.00–1.06	0.047
miR-216a-5p	1.03	1.00–1.06	0.047
miR-335-5p	0.98	0.97–1.00	0.048
miR-1260b	1.00	1.00–1.01	0.012

95% CI: confidence intervals.

## Data Availability

The datasets generated and analyzed during the current study are available on the repository Zenodo https://zenodo.org/records/10055012 (accessed on 20 January 2024).

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
