# Peer review of "Identification of a Panel of miRNAs Associated with Resistance to Palbociclib and Endocrine Therapy"

_ijms, 2024, doi:10.3390/ijms25031498_

Round 1
Reviewer 1 Report
Comments and Suggestions for Authors
The combination of anticancer treatments is a trend that lasts for quite some time. Indeed, this is one of the innovative ways to go, hence much research must be done. Due to the nature of the research, not all the results are positive. In this context, the manuscript has been evaluated since the treatment outcomes were more negative than positive. The manuscript is more interesting than just negative results because the authors tried to involve explanations like gene and expression pathway analysis, hence allowing to make a hypothesis. Also, one must state that clinical treatments have been done with more than 60 patients. This makes a negative result meaningful. Moreover, 7 miR’s was used. This could guide other research that will be conducted. To the reviewer’s mind, though the citation of this probable paper will be poor, it might be still worth publishing this data after intense corrections. The reviewer also underlines that the conclusions are very weak.
Major:
The tables are good, but their presentation of them is poor. Change commas to dots (points) in all the tables. The descriptions of Table 2 (probably they are at the end of table 2?) and Table 3 are too small. Please enlarge them. What are the differences in bold and not bold numbers in Table 3?
What is the figure at the end of the result sections? No naming no captions? The resolution is unacceptable. Part A states the predicted targets, but how the reader will understand this with this figure? The presentation to the least is uninformative. The B part is like a legend, which is understandable. I understood that the authors used KEGG and GO! For the analysis. But what is gene count? Strength FDRq value? Matching proteins with what? As this is the only positive part of the manuscript it must be well presented in order to publish this draft.
Also, why authors did not use REACTOME along with KEGG and GO! ? The reviewer would not ask this if the research data would be strong, hence it would be the choice of the authors. Now data is weak and only good bioinformatics can strengthen weak conclusions.
The text is written in acceptable English. However, there are a lot of layout errors. Two spaces, some citations are in squared brackets some of them are in simple ones. Not all sentences end in dots. And other errors...
As the main positive results can be from the bioinformatics side it’s common to have more than 28 citations.
In order for this paper to be published a lot of effort must be done in presentation to enhance the significance of this research.
Comments on the Quality of English Language
The text is written in acceptable English. However, there are a lot of layout errors. Two spaces, some citations are in squared brackets some of them are in simple ones. Not all sentences end in dots. And other errors...
Author Response
We thank the Reviewer for commenting this part of the results and for providing insightful suggestions. The tables are good, but their presentation of them is poor. Change commas to dots (points) in all the tables. The descriptions of Table 2 (probably they are at the end of table 2?) and Table 3 are too small. Please enlarge them. What are the differences in bold and not bold numbers in Table 3? The tables have been modified according to the reviewer’s comments . Table 3has been formatted differently and we hope that now the differences between bold and not bold numbers are detectable. What is the figure at the end of the result sections? No naming no captions? The resolution is unacceptable. Part A states the predicted targets, but how the reader will understand this with this figure? The presentation to the least is uninformative. The B part is like a legend, which is understandable. I understood that the authors used KEGG and GO! For the analysis. But what is gene count? Strength FDRq value? Matching proteins with what? As this is the only positive part of the manuscript it must be well presented in order to publish this draft. Also, why authors did not use REACTOME along with KEGG and GO! ? The reviewer would not ask this if the research data would be strong, hence it would be the choice of the authors. Now data is weak and only good bioinformatics can strengthen weak conclusions. As recommended, we improved the resolution of Figure 1 and we included Reactome analysis of potentially affected pathways. Further, we provide in Suppl. Table 1 the full list of miRNA targets used to generate the gene network represented in Figure 1A. the complete results of STRING analysis, in term of enriched pathways and functions, is now described in the new Suppl. Table 2. The text is written in acceptable English. However, there are a lot of layout errors. Two spaces, some citations are in squared brackets some of them are in simple ones. Not all sentences end in dots. And other errors... The text has been checked for lay out errors and we hope that now it is more readable. As the main positive results can be from the bioinformatics side it’s common to have more than 28 citations. The reference list has been implemented

Reviewer 2 Report
Comments and Suggestions for Authors
The manuscript titled "Identification of a panel of miRNAs associated with resistance to palbociclib and endocrine therapy" by Rosalba Torrisi and colleagues investigates the role of microRNAs (miRNAs) in the resistance to palbociclib, a CDK4/6 inhibitor, and endocrine therapy in hormone receptor-positive/HER2 negative metastatic breast cancer. The study involved evaluating miRNA profiles from tumor tissues of patients and identifying specific miRNAs that are statistically associated with progression-free survival (PFS). The authors propose a novel panel of miRNAs as potential biomarkers for predicting resistance to these therapies. This work provides some directions for future and potential research preliminarily. This manuscript contains certain clinical significance, minor revision is recommended before acceptance:
1. The image resolution of figure 1A and 1B must be improved.
2. Authors should clarify how they controlled for potential confounders and the statistical power of their analyses, given the sample size.
3. (Doi: 10.3736/jcim20050511) is recommend it to be added after “Stromal tumor-infiltrating lymphocytes (sTILs), defined as the percentage of lym-64 phocytes in the tumor stromal area, are the most reproducible immune parameter scored 65 by pathologists.”
4. The study's potential implications in clinical settings are practical. Authors should discuss how these findings could be integrated into current clinical practice and what further research is necessary before these miRNA panels can be used as predictive biomarkers.
5. Authors should discuss on how these limitations existed in this manuscript might affect the generalizability of the findings are needed.
6. Suggestions for future research, such as larger-scale studies or experimental validation of the miRNA function in resistance mechanisms, would strengthen the conclusion.
7. Authors should discuss the potential application of those miRNAs based on the Initial success of KNEYNOTE-942 clinical trial.
8. The conclusions of this manuscript should be further confirmed by in vitro or in vivo study

Author Response
Point-by-point response to the reviewer . First of all we thank the reviewer for his/her thoughtful and helpful comments. We tried to address the points raised adding new paragraphs to address the points 4-7 and modifying the conclusions according to his/her suggestions 1. The image resolution of figure 1A and 1B must be improved. The image has been improved and along with a more detailed reference we hope that now it is more understandable. 2. Authors should clarify how they controlled for potential confounders and the statistical power of their analyses, given the sample size. The following sentence has been added in the statistical method section: Considering the sample size and the estimated effect sizes, the power of analyses was not controlled. The results should be considered explorative in nature: the multivariable models are descriptive and should be considered as starting points for further analyses. 3. (Doi: 10.3736/jcim20050511) is recommend it to be added after “Stromal tumor-infiltrating lymphocytes (sTILs), defined as the percentage of lym-64 phocytes in the tumor stromal area, are the most reproducible immune parameter scored 65 by pathologists.” The recommended reference has been added. 4. The study's potential implications in clinical settings are practical. Authors should discuss how these findings could be integrated into current clinical practice and what further research is necessary before these miRNA panels can be used as predictive biomarkers. 5. Authors should discuss on how these limitations existed in this manuscript might affect the generalizability of the findings are needed. 6. Suggestions for future research, such as larger-scale studies or experimental validation of the miRNA function in resistance mechanisms, would strengthen the conclusion. 7. Authors should discuss the potential application of those miRNAs based on the Initial success of KNEYNOTE-942 clinical trial. 8. The conclusions of this manuscript should be further confirmed by in vitro or in vivo study Response to points 4-8 The following paragraphs have been added and the conclusions have been modified to address the points raised by the reviewer: A prospective validation of this panel of miRNAs, possibly assessed in liquid biopsies to overcome the need of metastatic tumor tissue, might represent an initial step to corroborate our findings, as well as in vitro studies. When validated, the identification of a panel of miRNAs predicting de novo resistance to treatment would allow to spare an ineffective treatment to a not negligible rate of patients (about 20%), who presumably may not derive any benefit from the standard 1st line treatment with ET and CDK 4/6i . In addition, in the era of mRNA-based vaccines as anti-cancer strategy (which may be fueled by the positive results of the trial KEYNOTE 942 despite in a much more antigenic tumor as melanoma) [32], the identification of miRNAs involved in response to treatment could help to develop specific mRNA directed vaccines to enhance CDK 4/6i activity. In conclusion, in the present study we propose a novel panel of miRNAs which predicted a higher likelihood of an early progression in patients treated with endocrine agents and the CDK 4/6i palbociclib. Due to the relative small number of patients our findings should be necessarily considered as exploratory and hypothesis- generating and deserve to be validated in larger cohorts and in in vitro models, but they may contribute to shed some light on the mechanism of de novo resistance to CDK4/6 i.

Round 2
Reviewer 1 Report
Comments and Suggestions for Authors
Almost all my concerns have been addressed except:
There are still very few citations.
And the resolution of the figure is still low.
Comments on the Quality of English LanguageBetter quality than before
Author Response
There are still very few citations. We added 3 additional references ( now total 38 vs 28) We think that we have included all the references relevant for the text, but we would be happy to consider any other specific reference recommended by the reviewer. And the resolution of the figure is still low. We did our best to improve the resolution of the figure and we attached 2 different formats of the figure (.jpeg and.pdf) and we hope that at least one of them succeeded to improve the figure
